# Making sense of unfamiliar COVID-19 vaccines: How national origin affects vaccination willingness

Eric A. Jensen[1], Brady Wagoner[1,2,3], Axel Pfleger[1]*, Lisa Herbig[1], Meike Watzlawik[1]

**1** Institute for Psychological Research at the SFU Berlin e. V., Berlin, Germany, **2** Department of Communication and Psychology, The Faculty of Humanities, Aalborg University, Aalborg, Denmark, **3** Bjørknes Høyskole, Oslo, Norway

* axel-pfleger@hotmail.com

**Data Availability Statement:** The datasets presented in this study can be found in online repositories. The names of the repository/ repositories and accession number(s) can be

## Abstract

Vaccination willingness is a critical factor in pandemics, including the COVID-19 crisis. Therefore, investigating underlying drivers of vaccination willingness/hesitancy is an essential social science contribution. The present study of German residents investigates the mental shortcuts people are using to make sense of unfamiliar vaccine options by examining vaccination willingness for different vaccines using an experimental design in a quantitative survey. German vaccines were preferred over equivalent foreign vaccines, and the favorability ratings of foreign countries where COVID-19 vaccines were developed correlated with the level of vaccination willingness for each vaccine. The patterns in vaccination willingness were more pronounced when the national origin was shown along with the vaccine manufacturer label. The study shows how non-scientific factors drive everyday decision-making about vaccination. Taking such social psychological and communication aspects into account in the design of vaccination campaigns would increase their effectiveness.

## Introduction

A deadly coronavirus does not know national borders, nor does it care about different flags, languages, or past conflicts. However, its host organisms–humans–care deeply about such things, and social biases, among other factors, have the potential to affect the rollout of a newly developed vaccine on multiple levels. In a context where few understand the intricacies of the technical differences between the different available vaccines, but a practical decision about whether to vaccinate needs to be made, the public needs to draw on other, non-scientific cues to fill in the gaps in information. Cues such as a vaccine's national origin can be used to develop attitudes about its quality and reliability, guided by perceptions of originating country. The influence of national perceptions is already apparent in the everyday naming of COVID-19 vaccines around the world. For example, the vaccine developed by Pfizer/BioNTech (Comirnaty) is largely being referred to as the "BioNTech" vaccine in Germany (with BioNTech being a German company), while in the USA it is mainly referred to as the "Pfizer" vaccine (with Pfizer being a US-American company).

found here: https://doi.org/10.5281/zenodo.5513261.

**Funding:** The research presented in this paper was funded by the German Federal Ministry of Education and Research under grant agreement 01KI20500. There was no additional external funding received for this study. The funders had no role in study design, data collection and analysis, decision to publish, or preparation of the manuscript.

**Competing interests:** The authors have declared that no competing interests exist.

People have the tendency to see nations as the natural, taken-for-granted state of the world and project their existence back into time immemorial. This is reinforced through everyday communication when, for example, talking about a certain vaccine, but also in weather maps, national celebrations, flags, football matches, and even the use of the pronouns such as 'we' and 'us' to refer to an 'imagined community' [1] of people that belong to the same nation. In reality, nationalism is "far from being an age-old 'primordial' condition, [but] has been produced by the age of the modern nation-state" [2 p9]. In its banal, taken-for granted form [2], nationalism is a key component of our everyday thinking. It is used to frame a variety of decision-making processes, including health decisions such as those around vaccination.

Medical crises are accompanied by some degree of uncertainty, which affects the practical decision-making people must take on for themselves and others to try to navigate such crises. Rational theorists [e.g., 3] dominate the early literature on health decision-making. These theorists propose that people make health decisions based on weighing the risks and benefits of a certain behavior. Until today, many interventions aiming to facilitate good health behaviors are based on those assumptions (e.g., through awareness-raising or information dissemination). However, research shows that the success of this approach is limited and often lacks the power to change people's behavior [e.g., 4, 5]. This also holds true for vaccine decision-making. Simply informing people about the benefits of a vaccine and the dangers of a disease such as COVID-19 is not enough to convince everyone to get vaccinated, despite the scientific consensus that approved vaccines are safe and effective [6, 7].

Dror and colleagues [8] were able to show that, indeed, the interplay of many factors influences the willingness to vaccinate against COVID-19. They collected data in Israel from 2470 physicians, life science graduates (biology, virology, chemistry, etc.), and from members of the general public without a life science background. They found that while the first physicians and science graduates based their reasoning on the technology underpinning the vaccine (e.g., mRNA), members of the general public concentrated more on the reported headline efficacy rate and the country of production. Here, the Israeli public preferred vaccines coming from the USA or UK over those from China or Russia despite a (hypothetical) 90% efficacy of a Chinese vaccine compared to 60% efficacy of a vaccine from the USA/UK. These results align with Israelis' attitudes about those foreign countries. Silver [9] shows that 82% of the people in Israel consider the USA to be their most reliable ally, while Russia and China received amongst the lowest ratings. Another survey conducted by Pew Research in Germany using nationally representative sampling from March to May 2021 (overlapping with the data collection period for the present study) found that the German public had a 62% favorability rating for the USA, and 63% favorability for the EU in general [10, 11]. German attitudes towards China and Russia were much less positive, with favorability ratings of 26% and 32%, respectively [12, 13].

It is thus possible that the above-mentioned 'imagined community' (in-group) and social representation of other nations (out-groups) can, among other factors, influence the willingness to get vaccinated with a certain product. The effect is mediated by an 'us'-feeling, because members of a group are more inclined to positive attitudes towards objects they are familiar with. This leads to positive evaluations and preference of one's own group, according to Mummendey et al. [14]. The development of specific attitudes toward the "others" (out-groups) only comes in a second subordinate step. Irrespective of this evaluation, however, there is a self-group bias, which functions as a projection of the individual onto the collective self by generalizing a typically positive self-image to the in-group. Outgroups cannot benefit from this generalization simply because they are "different" and are therefore evaluated less positively. Thus, an affective component has to be added in order to understand decision-making processes [e.g., 15]: People operate within a cognitive processing system (or rational system) and an affective system that operates more automatically and relies on emotions. Studies on a

then-hypothetical COVID-19 vaccine have already indicated that people tend to prefer domestic vaccines over foreign ones [16–18]). The results from Dror and colleagues [8] suggest that the in-group preferences may be extended to allies that are perceived as more familiar and therefore favorable, whereas out-groups are perceived as different, less familiar, and therefore less favorable.

No study so far has experimentally examined the influence of national origin on vaccination willingness in the post-approval phase. This study is designed to further explore the role of national origin from the perspective of citizens in Germany. We have formulated the following hypotheses:

**Hypothesis 1 (H1):** Germany's vaccines will attract higher levels of vaccination willingness than any other countries' (in-group preferences).

**Hypothesis 2 (H2):** The public in Germany will indicate higher levels of vaccination willingness for vaccines developed in countries that generally get more favorable ratings from them (extended in-group/allies) than those developed in countries that are perceived as "the other" (out-group).

**Hypothesis 3 (H3):** The pattern described in H2 will be more pronounced for those in the treatment group (seeing the national origin added to the vaccine label) than for the control group (which only sees the vaccine manufacturer name).

To evaluate the above hypotheses, we empirically examined the vaccination willingness of two randomly assigned groups: The first group was asked about their willingness to take a range of vaccines labelled by name only; the second group received the same question and response options, but was shown the national origin of each vaccine.

## Methods

The overall study protocol was approved by the Ethics Committee of the Sigmund Freud University.

### Survey design

This study was conducted as part of a national survey for the Viral Communication project (viralcomm.info). Respondents who had not been vaccinated against COVID-19 were initially asked to indicate whether they would voluntarily vaccinate against COVID-19, on a 5-point Likert-type scale with "Definitely not," "Probably not," "Maybe," "Probably," "Definitely," "Not applicable/No opinion," and "Prefer not to say" as the response options.

Those who selected "Maybe," "Probably," or "Definitely" were included in a subsequent posttest-only control group experiment with random group assignment, which is seen as a stable measure to identify cause-effect relationships [see 19, 20]. This particular experimental setup does not require pretesting as randomized grouping ensures probabilistic equivalence [19].

Using the same response options as above, both the control group and the treatment group were asked to indicate whether they would get vaccinated if they were offered a range of different COVID-19 vaccines. For each COVID-19 vaccine, the treatment group received the respective national origin for each vaccine as an additional piece of information in parentheses. For example, for "BioNTech/Pfizer" as shown to the control group, the treatment group was shown "BioNTech/Pfizer (German)". All respondents were asked about the following vaccines: BioNTech/Pfizer (German), Moderna (US-American), AstraZeneca (Swedish/British), CureVac (German), Johnson & Johnson (US-American), Sanofi/GSK (French), Sputnik V (Russian), and Sinovac (Chinese).

## Sampling and data management

Data were collected 2–22 March 2021 from a sample of the German population, aged 16 and above. Respondents were invited to participate who had previously taken part in a probability-based survey research project (end of 2020) and agreed to participate in further rounds of this study. Initial recruitment to the study was achieved by sending postcard invitations to a random selection of 30,000 households, using the German postal service's (Deutsche Post) address database. Addresses were stratified based on relative population size across German federal states [21].

To be included in the analysis, respondents were required to provide data for the following variables: age group, sex, nationality group (German/other), migration background, federal state, highest school leaving qualification, and highest professional qualification. These criteria were strictly required as weighting was applied next for the control group and treatment group using the latest available German census results [22]. Sample characteristics for all weighting questions were exactly aligned with the census. The final sample size was $N = 332$ ($\hat{p}_{woman} = 51\%$, $M_{age} = 48.2$, $SD = 17.2$ [weighted]).

## Data analysis

The Summer 2020 Survey Data by Pew Research Center [23] was used to calculate the valid proportions of China's and Russia's favorability ratings in Germany. Z-tests were performed to identify significant proportion differences between the country favorability ratings and the vaccination willingness related to the corresponding vaccines.

A related-samples Friedman test was used to identify significant differences in vaccination willingness between the different vaccines, and post-hoc Wilcoxon signed rank tests with Bonferroni correction were performed to test for significant pairwise differences. Mann-Whitney U tests were employed to identify significant differences between the control and treatment groups for each vaccine type. $\eta^2$ was calculated for each significant result to indicate the individual effect size. Proportions with 95% confidence intervals were ascertained for each response option, each respondent group (control and treatment), and each vaccine to display potential differences more clearly. Percentages were rounded to the nearest integer. Two-sided tests were conducted. Statistically significant results are reported at $\alpha < .05$ throughout this work.

## Results

The first step in this analysis was to compare the overall vaccination willingness results for each vaccine to assess whether there were statistically significant differences in willingness by vaccine. A Friedman test showed clear differences, $\chi^2(7) = 470.734$, $p < .001$, leading us to reject the null hypothesis of no differences in willingness between vaccines. Post-hoc Wilcoxon signed rank tests revealed that the German-developed BioNTech/Pfizer vaccine (generally known in Germany simply as "BioNTech") was strongly preferred over all other vaccines (see Table 1). The other Germany-based vaccine, CureVac (which was still in the clinical trials phase at the time of the survey), was preferred over Sanofi/GSK (French), Sputnik V (Russian), and Sinovac (Chinese).

This first analytic step confirmed H1, showing BioNTech/Pfizer, the German-originated vaccine currently in use, was preferred over all the others. Likewise, the other German-developed vaccine, CureVac, was preferred over other vaccines in the pre-approval stage. This indicates that a key driver here is nationalism, rather than perceptions of the objective superiority of the BioNTech/Pfizer vaccine over other options.

**Table 1. Pairwise Wilcoxon signed-rank comparisons with German COVID-19 vaccines for significant Friedman test.**

| Pairwise comparison | z | p | $\eta^2$ |
|---|---|---|---|
| BioNTech/Pfizer—Moderna | 5.485 | 0.000 | 0.19 |
| BioNTech/Pfizer—AstraZeneca | 7.763 | 0.000 | 0.39 |
| BioNTech/Pfizer—CureVac | 7.681 | 0.000 | 0.43 |
| BioNTech/Pfizer—Johnson & Johnson | 7.785 | 0.000 | 0.42 |
| BioNTech/Pfizer—Sanofi/GSK | 8.332 | 0.000 | 0.56 |
| BioNTech/Pfizer—Sputnik V | 9.571 | 0.000 | 0.66 |
| BioNTech/Pfizer—Sinovac | 9.469 | 0.000 | 0.68 |
| CureVac—Moderna | -4.994 | 0.000 | 0.18 |
| CureVac—AstraZeneca | 0.191 | 1.000 | |
| CureVac—Johnson & Johnson | -1.014 | 1.000 | |
| CureVac—Sanofi/GSK | 6.006 | 0.000 | 0.30 |
| CureVac—Sputnik V | 8.574 | 0.000 | 0.58 |
| CureVac—Sinovac | 8.667 | 0.000 | 0.60 |

P-values were adjusted with the Bonferroni correction.

The second step in the analysis was identifying whether the patterns in which vaccines were associated with higher levels of vaccination willingness aligned with the German public's existing general favorability ratings for countries outside of Germany. A Wilcoxon signed rank test comparing vaccination willingness between European/US-American vaccine origins (excluding Germany) and Russian/Chinese vaccine origins showed that vaccines with the former national origins were strongly preferred over those with the latter origins, $z = 9.482$, $p < .001$, $\eta^2 = .64$ (64% explained variance). This result aligns with German residents' positive rating of the USA and the EU in general (62% and 63% favorability, respectively) [10, 11] compared to the rather negative ratings of China and Russia (26% and 32%, respectively). In fact, the null hypothesis that the proportions of country favorability and vaccination willingness differ significantly was accepted respectively for China, $z = .047$, $p = .963$, and Russia, $z = .083$, $p = .934$. As Pew Research had not yet published the 2021 data, these tests could not be performed for the US-American and European vaccines (excluding those developed in Germany). However, the stand-alone favorability proportions and the difference in vaccination willingness give reason to believe they would result similarly.

The third analytic step was to investigate whether there were differences between treatment and control groups based on the embedded experimental design in which one group saw the vaccine manufacturer name only (control), and the other group also saw the national origin associated with that vaccine (treatment).

Statistically significant treatment effects were found for most vaccines, with national labels generally having the predicted effect (see Table 2). In general, the effect sizes were weak to moderate. Johnson & Johnson exhibited the strongest effect, followed by Sinovac and AstraZeneca. There was no significant shift for Sputnik V.

Greater vaccination willingness was identified when the national origin was made explicit for the following vaccines: BioNTech/Pfizer, AstraZeneca, CureVac, Johnson & Johnson, and Sanofi/GSK. Moderna and Sinovac attracted a lower vaccination willingness with the national origin made explicit. Table 3 shows the precise differences between the control and treatment groups for each of the response options. Considering the absence of negative responses for BioNTech/Pfizer, the increase in vaccination willingness for this vaccine was mainly restricted to the positive response options. Although there was a strong overall shift towards the extreme

**Table 2. Summary of Mann-Whitney U tests examining differences between the control and treatment group for each COVID-19 vaccine.**

| Vaccine | N | U | p | $\eta^2$ |
|---|---|---|---|---|
| BioNTech/Pfizer | 289 | 12070.000 | .001 | .04 |
| Moderna | 282 | 11134.500 | .026 | .02 |
| AstraZeneca | 253 | 9437.500 | .001 | .05 |
| CureVac | 230 | 7469.000 | .018 | .02 |
| Johnson & Johnson | 246 | 9608.500 | .000 | .08 |
| Sanofi/GSK | 198 | 5480.500 | .018 | .03 |
| Sputnik V | 248 | 6996.000 | .287 | |
| Sinovac | 200 | 3112.000 | .000 | .07 |

negative response option ("Definitely not") for Sinovac, there was also minor polarization towards the extreme positive response ("Definitely").

For both the control and treatment group, BioNTech/Pfizer and Moderna received the first and second highest proportions of people who would have "Probably" or "Definitely" gotten vaccinated, respectively. This proportion increased for Johnson & Johnson from the fourth to the third largest among all vaccines, while it dropped for Sinovac from the third to last to the last rank.

## Discussion

This study shows how scientific and public health issues such as COVID-19 vaccination are routinely filtered through an in-group and nationalist lens. Nationalism in particular is so widespread in contemporary culture as to pervade even a topic as seemingly technical as the safety and effectiveness of a vaccine for a disease driving a global pandemic. In particular, the present study focused on Germany, where the BioNTech/Pfizer vaccine was by far the most positively received in our study, with 98% and 99% vaccination willingness in the control and treatment group, respectively. This aligns with H1, supporting the hypothesis that in-group preferences and nationalism are drivers for attitudes towards vaccines and vaccination willingness. Our findings are consistent with in-group preferences and nationalism as explanations for divergent attitudes towards different vaccines, particularly when they are based on similar technologies and are similarly efficacious (as is the case with the two mRNA-based vaccines assessed here: BioNTech/Pfizer and Moderna). This trend is also evident when comparing two European vaccines in the pre-approval stage at the time of writing (CureVac and Sanofi/GSK).

Focusing on H2, people in Germany strongly favored vaccines with a European or USA origin over the Chinese and Russian vaccines. This is consistent with the highly favorable country ratings for the USA and the EU, compared to the low ratings for China and Russia. Strikingly, we found no significant differences between China's and Russia's favorability ratings, nor between the vaccination willingness for the vaccines developed in each of these countries. The confirmation of H2 suggests that in-group preferences and scientific nationalism not only apply to one's own country, but also to allied countries.

Regarding H3, we were able to confirm significant differences in vaccination willingness between respondents who were only shown the vaccine names (control group) and those who were additionally shown the vaccines' national origins (treatment group). BioNTech/Pfizer and Moderna were the preferred vaccines in both the control and the treatment group. However, vaccination willingness for BioNTech/Pfizer was significantly greater in the treatment group compared to the control group. This further supports an in-group and nationalism explanation for vaccination willingness for specific vaccines. Johnson & Johnson and

**Table 3. Summary of proportions for each group per COVID-19 vaccine, as well as the difference for each response option.**

| Vaccine | Response Option | Origin Not Explicit | | | Origin Explicit | | | $\Delta\hat{p}$ |
|---|---|---|---|---|---|---|---|---|
| | | $\hat{p}$ | 95% CI Lower Bound | 95% CI Upper Bound | $\hat{p}$ | 95% CI Lower Bound | 95% CI Upper Bound | |
| BioNTech/Pfizer | Def. not | 0% | 0% | 3% | 0% | 0% | 2% | 0% |
| | Prob. not | 0% | 0% | 3% | 0% | 0% | 2% | 0% |
| | Maybe | 2% | 0% | 6% | 1% | 0% | 4% | -1% |
| | Probably | 29% | 21% | 37% | 14% | 9% | 20% | -15% |
| | Definitely | 69% | 61% | 77% | 85% | 79% | 91% | 16% |
| Moderna | Def. not | 0% | 0% | 3% | 3% | 1% | 7% | 3% |
| | Prob. not | 8% | 4% | 14% | 3% | 1% | 7% | -5% |
| | Maybe | 6% | 3% | 12% | 12% | 7% | 18% | 5% |
| | Probably | 37% | 29% | 46% | 17% | 11% | 23% | -20% |
| | Definitely | 49% | 40% | 58% | 66% | 58% | 73% | 17% |
| Astra-Zeneca | Def. not | 29% | 21% | 39% | 8% | 4% | 14% | -21% |
| | Prob. not | 9% | 4% | 16% | 4% | 1% | 8% | -5% |
| | Maybe | 7% | 3% | 14% | 18% | 13% | 25% | 11% |
| | Probably | 22% | 14% | 31% | 27% | 20% | 34% | 5% |
| | Definitely | 32% | 24% | 42% | 43% | 35% | 51% | 11% |
| CureVac | Def. not | 14% | 8% | 23% | 3% | 1% | 8% | -11% |
| | Prob. not | 10% | 5% | 18% | 0% | 0% | 3% | -10% |
| | Maybe | 17% | 10% | 26% | 30% | 23% | 39% | 14% |
| | Probably | 28% | 19% | 38% | 25% | 18% | 34% | -2% |
| | Definitely | 31% | 22% | 42% | 41% | 33% | 50% | 10% |
| Johnson & Johnson | Def. not | 13% | 7% | 21% | 0% | 0% | 3% | -13% |
| | Prob. not | 17% | 10% | 25% | 6% | 3% | 11% | -11% |
| | Maybe | 15% | 9% | 24% | 17% | 11% | 24% | 2% |
| | Probably | 29% | 20% | 38% | 33% | 25% | 41% | 4% |
| | Definitely | 26% | 18% | 35% | 44% | 36% | 53% | 18% |
| Sanofi/GSK | Def. not | 8% | 3% | 16% | 7% | 3% | 13% | -1% |
| | Prob. not | 23% | 14% | 34% | 21% | 14% | 29% | -2% |
| | Maybe | 44% | 33% | 56% | 30% | 22% | 39% | -14% |
| | Probably | 18% | 10% | 28% | 15% | 9% | 22% | -3% |
| | Definitely | 7% | 3% | 16% | 27% | 20% | 36% | 20% |
| Sputnik V | Def. not | 24% | 17% | 33% | 35% | 27% | 43% | 11% |
| | Prob. not | 30% | 21% | 39% | 24% | 17% | 31% | -6% |
| | Maybe | 25% | 17% | 34% | 17% | 11% | 24% | -8% |
| | Probably | 11% | 6% | 18% | 15% | 9% | 22% | 4% |
| | Definitely | 11% | 6% | 18% | 10% | 6% | 17% | -1% |
| Sinovac | Def. not | 10% | 5% | 20% | 40% | 31% | 49% | 29% |
| | Prob. not | 26% | 16% | 37% | 21% | 14% | 29% | -5% |
| | Maybe | 31% | 21% | 43% | 19% | 13% | 27% | -12% |
| | Probably | 26% | 16% | 38% | 9% | 5% | 16% | -17% |
| | Definitely | 7% | 2% | 15% | 11% | 6% | 17% | 4% |

AstraZeneca vaccines showed the largest differences between treatment and control groups in vaccination willingness. Likewise, Sinovac received much lower vaccination willingness ratings amongst those for whom the national origin was made explicit. Overall, H3 was confirmed as well, as individuals' willingness to vaccinate was consistently greater for vaccines linked to 'in group' favored nations within the treatment group than the control group.

A limitation of this study affecting H3 is that national origins were probably already known to some respondents in the control group, rendering our experimental manipulation less strong than if the control group had been completely unaware of national origins. In such cases, the treatment is merely increasing salience of that national origin rather than introducing it for the first time. A likely implication of this limitation is that the treatment effects identified in this paper may be an underestimate.

Against the backdrop of a generally positive public mood internationally in the wake of the pandemic regarding science and its role in society [24], the news coverage of vaccines has focused on the latest research about the risks (e.g., blood clots) and benefits (e.g., efficacy rates). However, drivers for vaccine willingness are rarely so simple and rational. Vaccine decision-making happens within a complex system of interconnected components, such as the underpinning vaccine technology, vaccine delivery, and one's own background assumptions and viewpoints which is composed of various aspects (e.g., education, disease epidemiology, location within the social structure) [25]. Trust, in particular, functions as a mediator within vaccine decision-making [26].

Additionally, the impact of awareness of national origin on vaccination willingness might change over time. Another experimental study with a German-American sample conducted by Kobayashi and colleagues [27] in the vaccines' pre-approval phase could not find a state bias (tendency to prefer domestic vaccines over foreign ones). This contrast to our study could be an indicator that the effect only becomes apparent when various options are available. On the other hand, their experimental study only varied the national origin of the Pfizer/BioNTech vaccine as either "American" or "German", while through the extensive media coverage participants might already have been aware of the "double" national origin of the vaccine. Other studies on hypothetical COVID-19 vaccines in the pre-approval phase support our findings on national origin as a major factor in increasing/decreasing vaccination willingness [e.g., 16, 17]. It should be critically noted, however, that people in Germany are in the privileged position of being able to choose between different vaccines. That a national vaccine is among the choice options is also not a given. In countries where vaccines are scarce and difficult to access, and where national products are not available, the role of nationalism is certainly of a different nature. Further studies are needed to clarify which factors significantly influence vaccination decisions here.

Nevertheless, the present study contributes to the literature on vaccine willingness by uncovering the potentially powerful role of nationalism and other in-group biases in subtly influencing attitudes about vaccines. Not only could such attitudes affect vaccination rates in different countries, but they also affect the wider socio-political consensus about which vaccines should even be considered for use in each country. Results from the present study underscore just how ubiquitous in-group biases are.

The pattern identified here is by no means exclusive to the vaccine context or COVID-19 pandemic. In-group biases permeate socio-political discourse, providing people with a short-hand mechanism to identify who or what is trustworthy. As Douglas and Wildavsky [28 p9] pointed out, "people order their universe through social bias." These contexts are certainly used by the media to guide certain attitudes accordingly—a phenomenon that needs to be critically examined and reflected upon in order to better understand biases of certain world regions and the resulting consequences for global vaccination activities.

## Author Contributions

**Conceptualization:** Eric A. Jensen, Brady Wagoner.

**Data curation:** Axel Pfleger.

**Formal analysis:** Axel Pfleger.

**Funding acquisition:** Meike Watzlawik.

**Investigation:** Eric A. Jensen, Axel Pfleger.

**Methodology:** Eric A. Jensen, Brady Wagoner, Axel Pfleger, Lisa Herbig.

**Project administration:** Meike Watzlawik.

**Supervision:** Meike Watzlawik.

**Validation:** Eric A. Jensen.

**Writing – original draft:** Eric A. Jensen, Brady Wagoner, Axel Pfleger, Lisa Herbig.

**Writing – review & editing:** Eric A. Jensen, Brady Wagoner, Axel Pfleger, Lisa Herbig, Meike Watzlawik.

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
