## [Decision Letter · Decision Letter 0]

26 Oct 2021

PONE-D-21-25089Making sense of unfamiliar COVID-19 vaccines: How national origin affects vaccination willingnessPLOS ONE

Dear Dr. Pfleger,

Thank you for submitting your manuscript to PLOS ONE. After careful consideration, we feel that it has merit but does not fully meet PLOS ONE’s publication criteria as it currently stands. Therefore, we invite you to submit a revised version of the manuscript that addresses the points raised during the review process.

We look forward to receiving your revised manuscript.

Kind regards,

Shinya Tsuzuki, MD, MSc

Academic Editor

PLOS ONE

Journal Requirements:

"The funders had no role in study design, data collection and analysis, decision to publish, or preparation of the manuscript" 

Please include your amended Funding Statement within your cover letter. We will change the online submission form on your behalf

3. Please update your submission to use the PLOS LaTeX template. The template and more information on our requirements for LaTeX submissions can be found at http://journals.plos.org/plosone/s/latex

Additional Editor Comments (if provided):

I agree with the concerns pointed out by the reviewers then please respond to each of their comments.

Reviewers' comments:

Reviewer's Responses to Questions

**Comments to the Author**

1. Is the manuscript technically sound, and do the data support the conclusions?

Reviewer #1: Partly

Reviewer #2: Yes

2. Has the statistical analysis been performed appropriately and rigorously? 

Reviewer #1: Yes

Reviewer #2: Yes

3. Have the authors made all data underlying the findings in their manuscript fully available?

Reviewer #1: Yes

Reviewer #2: Yes

4. Is the manuscript presented in an intelligible fashion and written in standard English?

Reviewer #1: Yes

Reviewer #2: Yes

5. Review Comments to the Author

Reviewer #1: In the manuscript titled Making sense of unfamiliar 1 COVID-19 vaccines: How national origin affects vaccination willingness” the authors have shown the importance of nationalism in decision making for vaccine acceptance.

This is quite natural but not the only factor in decision making many other factors such as the technology used and vaccine efficacy, vaccine safety etc. are also involved in the decision making. While nationalism may be a factor but that is very region specific and may not be an important contributor for decision making.

Giving preference to the vaccine developed in Germany or USA rather than to Russia or China is very natural because the perception a population has about some countries in their overall dealing in geopolitical role, transparency in different national/international issue, human index etc. The better this factor are the more is the confidence in the product a country develops. Vaccines approval by WHO is also a major factor in decision making; Sputnik V, Chinese vaccine or even India Covaxin are not WHO approved which undermines the willingness of use

The study is very region specific also; the same question may be responded in a very different way in African region or in south east-Asian countries or country which don’t have vaccine candidate. This countries/region will make their decision based on the quality of vaccine and approval from WHO

In the above view my concerns are

1. Why the authors feel nationalism is an important factor in decision making for vaccine willingness and how important this factor is among the other factors (vaccine efficacy and safety)

2. How the decision making influenced by the global approval of a vaccine candidate

3. How the authors feel the nationalism factor will affect or change in decision making for under developed countries (to my opinion vaccine efficacy and safety will still be a decision making factor rather than nationalism or country of origin).

Comments highlighting the above issues need to be incorporated in the manuscript at appropriate palaces.

Reviewer #2: This study is highly relevant and explores an interesting question about how nationalism and in-group biases can influence people's willingness towards a vaccine. However, the cause for vaccine willingness is multifactorial and more acceptance for a specific vaccine candidate cannot be solely attributed to national origin. Additional data can be helpful to corroborate their conclusion further.

1) How many people in the study are aware of the vaccine efficacy? If so, nationalism may not be the determinant in higher acceptance for Pfizer/Moderna. This data can be included as a supplementary figure or added to the result section.

2) Line 262-263: Are there any references to validate this statement? The present study applies to Germany, but can this be extended to other countries, and what additional factors govern that?

Minor:

Table 1 and 2: Table titles are abbreviated.

6. PLOS authors have the option to publish the peer review history of their article (what does this mean?). If published, this will include your full peer review and any attached files.

Reviewer #1: No

Reviewer #2: No

---

## [Author Response · Author response to Decision Letter 0]

12 Nov 2021

Regarding the editor’s comments, we have added a statement about all research funding sources to the manuscript, including the indicated statement about additional external funding. The cover letter was amended to include the funded statement as well. 

Addressing the 1st comment by reviewer #1: 

We have added clarifications wherever it was appropriate that nationalism/in-group biases may be one of many factors influencing vaccine decision-making. However, the comment about other factors such as vaccine technology and efficacy does not apply as we did specifically control for the vaccines’ technology when testing hypothesis 1. Additionally, when testing hypothesis 3, we isolated the factor ‘vaccine origin’ by testing for differences in vaccination willingness within each vaccine (control group vs. treatment group), not across vaccines. Dror and colleagues [8] have also come to similar conclusions, as noted in the introduction, where we have added a small point for clarification of this. We additionally added clarifying prose in the discussion related to the aspect of similar technologies and efficacies. The last point we want to highlight regarding this comment is that in the discussion, we already explicitly stated that, “[v]accine decision-making happens within a complex system of interconnected components, such as the underpinning vaccine technology, vaccine delivery, and one’s own background assumptions and viewpoints which is composed of various aspects (e.g., education, disease epidemiology, location within the social structure) […]. Trust, in particular, functions as a mediator within vaccine decision-making […].” 

Addressing the 2nd comment by reviewer #1: 

In-group bias can very much arise from perceptions of individual countries’ socio-political circumstances. As an important example of in-group biases, we actively included different countries’ favourability ratings, and additionally tested the factor ‘nationalism’ among countries with similar socio-political circumstances as well as similar vaccine technologies and efficacies (Germany vs. France). 

Addressing the 3rd comment by reviewer #1: 

We investigated in-group preferences for COVID-19 vaccination in Germany and found that indeed, vaccine willingness is partially dependent on in-group biases. The results show that in Germany, vaccine willingness is skewed towards German vaccines and vaccines from socio-political allies. This of course does not mean that people in other countries will have the same in-group biases as in Germany. In our manuscript, we clearly refer to the observed biases as respective to Germany, e.g.: “This study is designed to further explore the role of national origin from the perspective of citizens in Germany.” We did, however, replace the last two sentences in the discussion to highlight this issue. On the topic of other factors influencing vaccine decision-making, please see our response to the first comment about controlling for such factors and isolating the factor ‘national origin’. 

Addressing the 1st comment by reviewer #2: 

We believe this comment has also been addressed by our reply to the first reviewer’s first comment. 

Addressing the 2nd comment by reviewer #2: 

We added the following prose before the referenced section in the comment: “It should be critically noted, however, that people in Germany are in the privileged position of being able to choose between different vaccines. That a national vaccine is among the choice options is also not a given. In countries where vaccines are scarce and difficult to access, and where national products are not available, the role of nationalism is certainly of a different nature. Further studies are needed to clarify which factors significantly influence vaccination decisions here.” 

Addressing the 3rd comment by reviewer #2: 

This appears to be a display error on the reviewer's computer or something similar. Otherwise, we do not know what the reviewer is referring to in this comment. There are no abbreviations in the table titles.

---

## [Decision Letter · Decision Letter 1]

26 Nov 2021

Making sense of unfamiliar COVID-19 vaccines: How national origin affects vaccination willingness

PONE-D-21-25089R1

Dear Dr. Pfleger,

We’re pleased to inform you that your manuscript has been judged scientifically suitable for publication and will be formally accepted for publication once it meets all outstanding technical requirements.

Kind regards,

Shinya Tsuzuki, MD, MSc

Academic Editor

PLOS ONE

Additional Editor Comments (optional):

Reviewers' comments:

Reviewer's Responses to Questions

**Comments to the Author**

1. If the authors have adequately addressed your comments raised in a previous round of review and you feel that this manuscript is now acceptable for publication, you may indicate that here to bypass the “Comments to the Author” section, enter your conflict of interest statement in the “Confidential to Editor” section, and submit your "Accept" recommendation.

Reviewer #1: All comments have been addressed

Reviewer #2: All comments have been addressed

2. Is the manuscript technically sound, and do the data support the conclusions?

Reviewer #1: Yes

Reviewer #2: Yes

3. Has the statistical analysis been performed appropriately and rigorously? 

Reviewer #1: Yes

Reviewer #2: I Don't Know

4. Have the authors made all data underlying the findings in their manuscript fully available?

Reviewer #1: Yes

Reviewer #2: Yes

5. Is the manuscript presented in an intelligible fashion and written in standard English?

Reviewer #1: Yes

Reviewer #2: Yes

6. Review Comments to the Author

Reviewer #1: The authors have done the needful changes to reflect that the outcome of the study is specific to geo-political location and available freedom for making choices for vaccine and the criteria for choice will be very different depending on the available privilege to other country.

Reviewer #2: (No Response)

7. PLOS authors have the option to publish the peer review history of their article (what does this mean?). If published, this will include your full peer review and any attached files.

Reviewer #1: No

Reviewer #2: No

---

## [Editor Report · Acceptance letter]

6 Dec 2021

PONE-D-21-25089R1 

Making sense of unfamiliar COVID-19 vaccines: How national origin affects vaccination willingness 

Dear Dr. Pfleger:

I'm pleased to inform you that your manuscript has been deemed suitable for publication in PLOS ONE. Congratulations! Your manuscript is now with our production department. 

Kind regards, 

on behalf of

Dr. Shinya Tsuzuki 

Academic Editor

PLOS ONE